# How Glaciers Function and How They Create Landforms: Testing the Effectiveness of Fieldwork on Students' Mental Models—A Case Study from the Sanabria Lake (NW Spain)

Diego Corrochano [1,*] and Alejandro Gómez-Gonçalves [2]

1   Department of Mathematics Education and Experimental Sciences Education, University of Salamanca, 37008 Salamanca, Spain
2   Department of Geography, University of Salamanca, 37008 Salamanca, Spain; algomez@usal.es
*   Correspondence: dcf@usal.es

**Abstract:** This paper analyzes the impact of fieldwork on the development of students' mental models concerning glaciers and their effects on the landscape. Data were collected by means of an open-ended questionnaire that was administered to 279 pre-service teachers before and after an educational field trip, which analyzed its impact on short-term and long-term outcomes. In general, students' mental models about how glaciers function and how they create landforms are relatively simplistic and incomplete. Students are unaware of the major erosional properties associated with glaciers and many of them do not specify that glaciers are bodies of ice that have a tendency to move down slope. The analysis of the data yielded four mental model categories. Fieldwork influenced the short-term effects on mental model development even though its positive impact decreases over time. Mental models including scientific views were only found in the post-instruction group. On the other hand, the pre-instruction group was strongly influenced by a catastrophic event that occurred in the region in 1959 (the Ribadelago flooding), which interferes with students' mental reasoning on the formation of landscape features. This way of thinking is reinforced and/or mixed with a religious myth (Villaverde de Lucerna legend), which also invokes a catastrophic origin of the lake. In this case, this includes mystic flooding.

**Keywords:** fieldwork; mental model; glaciers; mental representation; alternative conceptions; misconceptions; field trip; surficial processes; geosciences education; geography education

## 1. Introduction

Earth Sciences differ from other scientific disciplines in that most of the key natural phenomena cannot be directly observed from the temporal and spatial perspective of human experience. Therefore, students are required to build up their own mental models to represent their views of some Earth system processes and features. However, given the abstract and complex nature of these concepts, knowledge acquisition cannot be accomplished solely through direct perception and individual construction. Knowledge must be acquired with the help of explanations and exposure to scientific and culturally accepted information [1]. In fact, mental models are dynamic and are always under construction based on new knowledge, ideas, conceptions, and personal experiences [2]. When exposed to educational interventions, students are supposed to develop or change their views to more scientific ideas. Knowing students' ideas helps teachers and educators design the scientific curricula and types of instruction that can best modify such alternative conceptions and strengthen scientific ones [3,4].

Over the years, Geography, Environmental Sciences, and Earth Sciences have complemented traditional classroom teaching with fieldwork or educational field trips as essential instructional components of these disciplines. However, empirical research on the effectiveness and impact of different fieldwork approaches is sparse in the existing literature and the production of studies in the area has only recently begun [5].

Two learning activities have been identified during field trips [6]: affective and cognitive. The affective aspect has been less explored because of its relation to emotions, feelings, and values. During fieldwork, students demonstrated significant affective responses, which have been suggested to encourage deep learning [6]. On the other hand, more attention has been paid to the importance of fieldwork as a key teaching strategy to improve the construction of scientific knowledge (see review in Reference [7]). For example, some authors have proven that students who complete field-based introductory science courses have a much better understanding of geoscience concepts than students whose introductory geoscience course did not involve any sort of field experience [8]. Nevertheless, and despite this, because of the logistical difficulties that challenge data collection over long periods of time, few researchers have studied the long-term impact of field trips [7].

In order to enhance field-based education and try to bridge the research gap found in the literature, this paper analyzes the impact of fieldwork on the development of students' mental models about surficial processes, landscape formation, glaciers, and climate changes. Specifically, the purposes of this study are: 1) to identify and characterize the different types of mental models held by teacher training university students about the formation of a glacial lake (Sanabria Lake, NW Spain), and 2) to test the short and long-term one-day fieldwork impact on the evolution of such models.

*1.1. Misconceptions and Mental Models in Earth Sciences*

To understand the world and its natural processes, students build up spontaneous internal representations or mental models that are useful to make predictions or explain phenomena or events [2]. These representations, developed by the human mind, can be expressed through action, speech, writing, or drawing [8]. Since mental models operate in the abstract, the learning of scientific concepts, given their high degree of abstraction and complexity, require the construction of such models [9]. Learning science consists of the need for conceptual change. According to the ideas of some authors, conceptual change occurs when the ontological category to which a concept is assigned radically changes [10]. Other authors suggest that this process is gradual and slow, and occurs when students encountered new scientific information [1,11]. Based on these authors, students change only some of the ideas and beliefs of their framework theory while the others remain, giving rise to intermediate synthetic models. Either way, adequate acquisition of scientific concepts requires a fundamental reconstruction of students' previous mental models to build new conceptual ones [2,12], which would be external models adopted by expert groups as accurate and reasonable representations of natural phenomena [2]. Extensive research has shown that students come to the classroom with prior ideas or alternative frameworks that can be structured into mental models [13]. In many cases, these ideas are misconceptions that may prevent appropriate learning of the studied concepts [14]. It has been suggested that, once embedded in a conceptual framework, these conceptions are highly persistent and resistant to change, which can affect subsequent learning [15]. Misconceptions abound in many scientific disciplines (see review in Reference [16]). Although, there has been comparatively less research on this issue in the area of geosciences than in other sciences [17,18]. Moreover, rather than considering a particular intervention, most of the studies that have addressed geoscience conceptions are descriptive [19]. The geoscience misconceptions reported in the literature are mostly related to plate tectonics, earthquakes, historical geology, weathering and erosion, rocks and minerals, earth structure, soil, geologic resources, and volcanoes [5,19–21]. Nevertheless, there is not much research into students' ideas about the role of surficial processes in the formation of landscape and relief. Likewise, research focused on these ideas should be centered on how students' thinking changes over time [22]. To date, studies on these processes mostly dealt with weathering and erosion, river formation, and the water

cycle [22–24]. Specifically, there are only a few noteworthy studies that address students' understanding of glaciations and their landscape effects [25–30]. There are also other instances of research on glacial activity that have focused on internal movement mechanisms, erosion and deposition, and on the relative timing of glacial advances and retreats, but have not been published in peer-reviewed journals [20]. Therefore, there are only a few recent articles about students' understanding of how glaciers are formed and the types of landforms they create. Therefore, there is a considerable lack of research examining students' ideas of glaciers [27]. This is surprising, given the attention devoted to climate change and its impact on the retreat and melting of glaciers and ice sheets. One of these studies is focused on the understanding of glacial function and found that, while elementary students realize that glaciers are bodies of ice that have a tendency to move, in general, they did not frame glaciers within a perspective of fluctuating climate, and neither were they aware of the major erosional properties associated with them [25]. In contrast, other research studies have pointed out that students commonly have difficulties to understand that glaciers are dynamic systems with a continuous flow of ice and conceive them as solid bodies of ice that do not move [26,27]. Other conceptions contemplate fluid water as the moving agent of the glacier: "within a cyclic process, glacial ice melts, moves as fluid water, refreezes, melts again, and then moves further" [26]. Some other studies pointed to the importance to master the main events across geological time in order to understand the Ice Age [29,30] and how glacier landforms are created such as U-shaped valleys [28].

### 1.2. Fieldwork and Its Educational Impact

School field trips or fieldwork can be defined as a curriculum-related activity that involves venturing outside the four walls of the classroom to learn from first-hand experience [6,31]. Educators use field trips to develop the relationship between students and their local environment [32] and to reinforce what has been previously taught in class [33], letting facts speak for themselves and evoking senses and feelings that are difficult to introduce as part of the regular classroom routine [34]. In this regard, outdoor areas become key teaching resources that foster active, participative, and meaningful learning through educational situations that would be hard to provide in the classroom [35], such as the study of natural processes and their results on their various scales, dimensions, and levels of complexity, the application of open-air research skills and techniques, and the usage of a wide range of new problem-solving and research possibilities [5].

There are certain factors that play a key role in school field trips [7]: a) innovation of the trip environment, b) social interaction during the trip, c) social context of the trip, d) previous knowledge and conceptions of a topic, and e) personal interest and motivation. As mentioned above, while there are certain studies that provide an analysis of the impact of fieldwork on the construction of scientific knowledge [35–38], there are only a few that have tried to measure the effectiveness of field trips on students' mental models and their misconceptions. For example, a study on students' mental models of desert environments concluded that they did not change significantly after the field trip [39]. According to such research, the reason for this could be that changing mental models was not among the learning goals of the experience. Against this, other studies demonstrate that field trips encourage significant changes in students' mental models and foster better understanding of local environments [40,41]. These studies claim that field trips have encouraged students to become actively responsible, which also deepens their understanding of local environments.

## 2. Geological and Geographical Context of the Sanabria Lake

The Sanabria Lake (42°07′30″N, 06°43′00″W; 1000 m a.s.l.) is located in the province of Zamora (NW Spain), on the eastern slope of the Trevinca Massif, which is a mid-latitude mountain range that is currently ice-free. It is the largest glacial lake in the Iberian Peninsula (368 ha), with a maximum length of 3.16 km W to E, a maximum width of 1.53 km, and a volume of 96 Hm$^3$. The bottom of the lake is relatively flat corresponding to a typical U-shaped glacial valley. The lake bathymetry exhibits two over-excavated sub-basins including the western one (46 m maximum depth) and the eastern one (51 m maximum depth),

which are separated by a ridge that is 20 m below the water surface [42]. The lake is warm (4.5–24.8 °C water temperature) and oligotrophic, thermally stratified from March to November [43]. The Tera river is responsible for the lake's main inflow and outflow of water.

The Sanabria Lake is geologically located in the "Ollo de Sapo" anti-form, which is a tectonic structure that separates the West Asturian-Leonese Zone and the Central-Iberian Zone from the Hercynian Iberian Massif. The lake is situated between the mountains of the Sierra Segundera and the Sierra Cabrera, characterized by their metamorphic or plutonic felsic rocks composed of gneisses and granodiorites. The lake stretches over a glacial depression in the Tera valley and is dammed by a main terminal moraine complex on its western side. The depression is the result of glacial erosion during the Würm glacial period, when outlet valley glaciers drained radially from a large plateau ice cap centered on the Sierra Segundera that covered an area of over 440 km$^2$, with ca. 300 m maximum ice thickness and outlet glaciers reaching as low as 1000 m [44]. Two main glacial arms descended from the Sierra Segundera along the Tera and the Segundera and Cardena valley ensemble, coming together in the lower part of the Tera valley, which is the current location of Sanabria Lake. Several glaciation and deglaciation episodes have been reported [44–46] even though a rapid final melting of the ice seems to have occurred in 11.7 to 10.1 ka BP followed by a period of high river discharge from 10.1 to 8.2 ka BP [46].

In 1959, the region suffered a tragic disaster. A small dam called Vega de Tera, located approximately 3 km up valley of the lake, failed during the night of 9 January, flattening the village of Ribadelago and killing 144 villagers. This flooding of 1959 is usually confused with a colorful medieval legend that forebode the catastrophe. According to such a legend, at the bottom of the lake lies a village (Villaverde de Lucerna) that was sunk a long time ago, when Jesus, dressed as a pauper, arrived begging alms and was turned away by the villagers. He then raised his staff and drove it into the ground, commanding water to rise from the hole, and out it gushed, flooding the village, drowning all its people, and leading to the formation of the lake. Such a legend became popular thanks to Miguel de Unamuno, a famous Spanish writer and philosopher, who mentioned it in his 1931 novel "San Manuel Bueno, mártir."

## 3. Method

### 3.1. Participants

The sample was made up of 279 volunteer pre-service teachers (209 women and 75%, 79 men and 25%, with an average age of 21.1, SD 2.9) taking a Teacher Training Degree at a medium-size public university in North-Eestern Spain. The sample included 85 first-year students (Group 1, GR1), 84 second-year students (Group 2, GR2), 50 third-year students (Group 3, GR3), and 60 fourth-year students (Group 4, GR4). At the end of the second semester of the second year, students were taken on a field trip to the Sanabria Lake Natural Park, so that, at the time of the study, GR2 had carried out the fieldwork two weeks earlier, group 3 one year earlier, group 4 two years earlier, and GR1 had not yet performed it. The fieldwork was supervised by the same teachers (researchers) every year. It was proposed as part of Natural and Social Sciences Education subjects and mostly involved landscape interpretation and the analysis of different elements related to rocks, relief, and vegetation.

### 3.2. Instrument and Procedure

The study was conducted in spring 2018, at the end of the second semester. A short paper-based questionnaire of six *ad-hoc* open-ended questions was designed and delivered to identify students' mental models of glacial formation (see Tables 1 and 2). Therefore, the mental models revealed by the study are limited to the participants' written answers. The goals and limitations of writing as a means of communicating mental ideas are similar to those of verbal communication in interviews or communication through drawings. Despite such limitations, open-ended questionnaires have been proven to be useful tools for analyzing mental models in previous studies [47–50].

The introductory heading of the questionnaire includes an explanation clarifying that the survey was focused on students' perceptions and ideas about the origin of the Sanabria Lake, which is the largest lake of glacial origin of the Iberian Peninsula. Students knew that the lake was of glacial origin before starting the questionnaire and should have answered it from a glacial perspective. The content of the questionnaire was developed based on misconceptions documented in the research literature [20,25–27] or encountered during previous instruction. Open-ended generative questions were used to obtain further information and a comprehensive view of students' mental models. Students answered the questionnaire individually and there was no established time-limit. During this time, students were able to switch back to previous questions to change their answers if they wished. Emphasis was placed on the fact that what was being asked was each student's personal thoughts, regardless of whether responses were "scientifically correct" or not. The questionnaire was administered to four different student groups: GR1 (pre-intervention) and GR2-GR3-GR4 (post-intervention). Two PhD experts (a geologist and a science education expert) reviewed and assessed its questions in order to verify their appropriateness for eliciting responses involving students' ideas about the origin of the lake.

The open-ended nature of the questionnaire required an inductive approach, since students' thoughts were described in words. Therefore, the categories were inductively generated from the students' answers. Instead of including the answers in predetermined categories, the latter were generated when reading and interpreting the data. A first reading of the questionnaires led to the definition of several categories according to key ideas born from the interpretation of each answer. Similar responses were grouped even though certain long answers were subdivided and included under two or more different categories. Categories were also added or redefined to accommodate new data. When in doubt, answers were examined several times by both researchers until an agreement on the appropriate categories was reached. This first phase of the analysis provided insight on the frequency of students' key ideas when answering each question. During the next phase of analysis, categories were grouped into wider typologies that reflected students' mental models to subsequently analyze the distribution of mental models across different student groups. Pearson chi-square ($\chi^2$) statistical tests were used to determine statistical significance of the frequencies of the mental models between groups and from pre-intervention to post-intervention. Chi-square tests were conducted using Microsoft Excel and were determined by using 2×4 matrices. The frequency of the mental models across GR1 (pre-intervention) and GR2-GR3-GR4 (post-intervention) was compared to establish the impact of fieldwork on mental models. In addition, the frequency of the mental models across GR2 and GR4 was compared to determine the short-term and long-term impact of fieldwork on the development of mental models.

### 3.3. Brief Description of the Educational Field Trip

The field trip was carried out through the western side of the province of Zamora Province (NW Spain) and included activities in the Natural Park of the Sanabria Lake. According to research logic, and so that it could be integrated into the curriculum, fieldwork was organized as follows [35,51]: a) a previous preparatory one-hour session in the classroom, where students took a previous lecture about the trip and the formation of glacial landforms, b) field trip according to the designed itinerary [52], and c) briefing after the trip to include students' ideas and perceptions. The purpose of this plan was to increase the influence of fieldwork on learning by linking the trip to a regular classroom routine.

The activities carried out in the Sanabria Lake Natural Park consisted of: a) a visit to the Natural Park's interpretive center, where students analyzed audio-visual presentations, physical models of the lake (mockup), maps, and other material, b) guided observation of glacial landforms from the San Martín de Castañeda lookout, where students drew a geomorphological sketch of the landscape and the Sanabria Lake and its moraines, c) a three-hour walk across the ancient glacier's north lateral moraine to the Tera valley, where students identified and analyzed some glacial forms such as depositional landforms (moraines, glacial erratics … ) or erosional landforms (groves and striations, U-shaped Tera valley … ) on their own. This fieldwork had an integrated inquiry-based design [53], which means that it

was student-centered and teachers played only a minor role in terms of talking, where their interventions were limited to asking inquiry-based questions to support students' investigation and reasoning.

All pre-service teachers of this investigation have completed the field trip to the Sanabria Lake Natural Park at the end of the first semester in their second university course. Previously, during this same course, they have studied two subjects where they have analysed the landscape and landforms of Natural Sciences Education and Social Sciences Education. Students have learned about glaciers and about how they function, focusing their attention on glacial landforms such as glacial lakes, moraines, or U-shaped valleys. After these two courses in their second university course, students do not receive further instruction about this topic.

## 4. Results

Students' understanding of glacial processes and landforms has been arranged into different categories based on the inductive analysis of the answers to each question (Tables 1 and 2). Besides, certain categories have also been subdivided into subcategories (e.g., category 2A has been subdivided into subcategories 2A1, 2A2, and 2A3).

**Table 1.** Categories and subcategories of students' ideas about glaciers and their effects (I of II).

| Q1: Try to explain, in your own words, how the Sanabria Lake was formed: what processes intervened in its formation? | TOTAL (n = 279) % | GR. 1 (n = 85) % | GR. 2 (n = 84) % | GR. 3 (n = 50) % | GR. 4 (n = 60) % |
|---|---|---|---|---|---|
| 1A.1: Glacial motion + two other combined processes | 8.6 | 1.2 | 15.5 | 10.0 | 8.3 |
| 1A.2: Glacial motion and other combined processes | 2.5 | - | 4.8 | 2.0 | 3.3 |
| 1B.1: three or more combined processes (no glacial motion) | 6.1 | 2.4 | 8.3 | 4.0 | 10.0 |
| 1B.2: two combined processes (no glacial motion) | 7.9 | 7.1 | 7.1 | 6 | 11.7 |
| 1C.1: Glacial ablation (melting ice) | 33.7 | 29.4 | 38.1 | 34.0 | 33.3 |
| 1C.2: Glacial ablation under warmer conditions | 3.9 | 4.7 | 1.2 | 10.0 | 1.7 |
| D. Something occurred in a glacier or during or after a glaciation | 21.9 | 11.8 | 19.0 | 34.0 | 30.0 |
| E. Ribadelago disaster (breakage of the Vega del Tera dam) | 5.4 | 17.6 | - | - | - |
| 1F.1: Tectonics, river erosion, rain, etc. | 6.1 | 14.1 | 6.0 | - | - |
| 1F.2: No sense or not answered | 3.9 | 11.8 | - | - | 1.7 |
| **Q2: Why do you think that scientists claim that the Sanabria Lake is of glacial origin? What evidence do you think they have to affirm that?** | | | | | |
| 2A.1: both types of landforms (depositional and erosional) naming several (three or more) of them: moraines, erratics, U-shaped valleys … | 4.3 | - | 10.7 | 2.0 | 3.3 |
| 2A.2: both types of landforms naming one or two of them | 9.7 | 2.4 | 19.0 | 8.0 | 8.3 |
| 2A.3: both types of landforms without any term | 3.6 | 1.2 | 3.6 | 10.0 | 1.7 |
| 2B.1: one type of landform (depositional or erosional landforms) and its term | 20.4 | 10.6 | 17.9 | 18.0 | 40.0 |
| 2B.2: one type of landform without any term | 18.6 | 7.1 | 22.6 | 26.0 | 23.3 |
| 2C.1: Chemical or physical properties of the water | 12.2 | 22.4 | 3.6 | 18.0 | 5.0 |
| 2C.2: No sense or not answered | 31.2 | 56.5 | 22.6 | 18.0 | 18.3 |

Answers to question 1, about the processes involved in the lake's formation, have been grouped into six different categories (1A–1F). Category 1A comprises all the answers that suggest glacial motion or transport (intrinsically express movement) and other combined processes, such as erosion, sedimentation, ablation, and more. In turn, category 1B includes answers that, while mentioning two or three combined processes, make no reference to glacial motion. Category 1C includes the idea that

the lake formed just after the melting of ice, without mentioning the creation of the erosive basin. Category 1D encompasses those answers that only mention glaciation or factors that would have occurred during a glaciation as the main process related to the lake's formation. Category 1E refers to the breakdown of the Vega del Tera dam and the subsequent flooding of the village of Ribadelago. Lastly, Category 1F includes other answers (e.g., lake formation as the result of tectonic movements, abundant rain, etc), meaningless answers, and unanswered questions.

Question 2 is about glacial landforms (scientific evidence supporting the glacial origin of the lake). Answers have been grouped into three categories (2A–2C). Category 2A includes all the answers that mention both types of glacial landforms (erosive and depositional), whereas Category 2B includes those answers referring only to one type of landform (erosive or depositional). Category 2C is for other answers, meaningless answers, and unanswered questions.

**Table 2.** Categories and subcategories of students' ideas about glaciers and their effects (II of II).

| Q3: Do you know approximately when the lake was formed? | TOTAL (n = 279) % | GR. 1 (n = 85) % | GR. 2 (n = 84) % | GR. 3 (n = 50) % | GR. 4 (n = 60) % |
|---|---|---|---|---|---|
| 3A.1: 100.000 to 1.000 years ago | 30.1 | 10.6 | 40.5 | 48.0 | 28.3 |
| 3A.2: thousands of years ago | 8.2 | 1.2 | 9.5 | 4.0 | 20.0 |
| 3A.3: Quaternary or last glaciation | 7.2 | 9.4 | 9.5 | 4.0 | 3.3 |
| 3B.1: millions of years ago | 9.3 | 8.2 | 8.3 | 8.0 | 13.3 |
| 3B.2: hundreds of years ago (quantitative or qualitative data) | 5.7 | 11.8 | 6.0 | 2.0 | - |
| 3B.3: when the dam of Vega del Tera broke down (50-75 years ago) | 5.7 | 18.8 | - | - | - |
| 3C: no sense, not answered, or not quantified | 33.7 | 40.0 | 26.2 | 34.0 | 35.0 |
| **Q4. Do you know approximately how long it took for the lake to form?** | | | | | |
| 4A.1: 100,000 to 1,000 years | 8.6 | 1.2 | 18.8 | 6.0 | 6.7 |
| 4A.2: thousands of years | 9.0 | 4.7 | 16.5 | 4.0 | 8.3 |
| 4B.1: >100,001 years (even up to Myr ago) | 5.7 | 4.7 | 4.7 | 2.0 | 11.7 |
| 4B.2: hundreds of years (quantitative or qualitative data) | 6.8 | 9.4 | 3.5 | 8.0 | 6.7 |
| 4B.3: 1-10 years | 2.9 | 8.2 | - | 2.0 | - |
| 4C: no sense, not answered, or not quantified | 67.0 | 71.8 | 55.3 | 78.0 | 66.7 |
| **Q5. From your point of view, could there be a submerged village at the bottom of the Sanabria Lake? Why do you think that?** | | | | | |
| 5A: no, it is not possible because no human settlement could have existed | 59.9 | 29.4 | 81.0 | 82.0 | 55.4 |
| 5B.1: yes, it could be possible (mostly referred to the Vega del Tera catastrophe) | 32.6 | 58.8 | 11.9 | 14.0 | 40.4 |
| 5B.2: do not know, uncertain, or not answered | 7.5 | 11.8 | 7.1 | 4.0 | 5.0 |
| **Q6. Why do you think the water of the lake is so cold?** | | | | | |
| 6A: due to the cool climate (other explanations, such as great water depth could be also suggested) | 32.3 | 28.2 | 38.1 | 38.0 | 25.0 |
| 6B: due to great water depth and/or vertical water circulation | 5.7 | - | 3.6 | 16.0 | 8.3 |
| 6C: water comes from melting ice: glacial water (additional explanations could be also suggested) | 55.9 | 64.7 | 50.0 | 44.0 | 61.7 |
| 6D: other or not answered | 6.1 | 7.1 | 8.3 | 2.0 | 5.0 |

Answers to question 3, concerning the time of the lake's formation, have been grouped into three categories (3A, 3B, and 3C). Category 3A includes those answers that properly quantified the timing (absolute and relative ages) of lake formation. Category 3B includes those answers that indicated ages too far above or too far below the scientifically accepted, and also those referring to the date when the Vega del Tera dam failed (3B3). Category 3C comprises other answers, meaningless answers, and unanswered questions.

Question 4 refers to the approximate duration of the lake's formation. Answers have been grouped into three categories (4A, 4B, and 4C). In a similar way as with question 3, Category 4A includes those answers that properly quantified the approximate duration of the lake's formation and Category 4B indicated much more or much less time than the scientifically accepted duration. Category 4C comprises other answers, meaningless answers, and unanswered questions.

Answers to question 5, about the possible existence of a submerged village at the bottom of the lake, have been grouped into two categories (5A and 5B). Category 5A includes those answers claiming that it is impossible because no human settlement could have existed during glaciation. Category 5B includes those answers that state that it is possible (5B1) include meaningless answers and unanswered questions (5B2).

Lastly, answers to question 6, asking why the lake's water is so cold, were divided into four categories (6A, 6B, 6C, and 6D). Category 6A includes answers that affirm that the water is so cold because of the area's climate. Additional explanations mentioning great water depth, or the melting of the snow of its surrounding snow-capped mountains every spring could be suggested. Category 6B includes those answers attributing the lake's low water temperature only to great depth or to vertical movements and water circulation that would prevent water warming. Category 6C includes those answers that claim that water is so cold because it comes from melting ice, which means primogenial glacial water. Category 6D comprises other answers, meaningless answers, and unanswered questions.

After describing and analyzing the categories and subcategories established for each question, all of them were grouped into four main mental models to explain the lake's formation and glacial functioning (Figure 1). The questions about how the lake was formed and which processes intervened in its formation, and about if could there be a submerged village at the bottom of the Sanabria Lake, are considered key to discover students' understating and reasoning processes on the issue of glacial landform formation and development. The information of the other answers (questions 2, 3, 4, and 6) was used to complete and to check the internal coherence of mental models. The data were subsequently re-analyzed for confirmation and to resolve disagreements. However, while mental models are not intended to represent a conceptual hierarchy, it is true that the dynamic mental model is conceptually more complex and better reflects the scientific or conceptual model of lake formation. The four mental models defined for the participating students are described below.

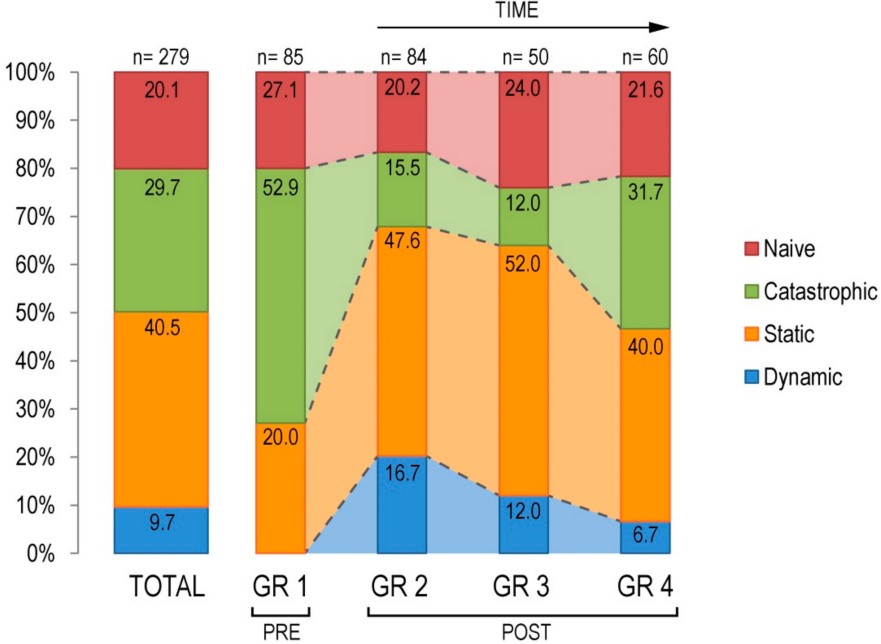

**Figure 1.** Identified mental models of glaciers and their effects, across GR1, GR2, GR3, and GR4, and all students. Note that the dynamic model is only present in post-intervention groups and how its presence is reduced over time.

*a) Dynamic mental model:* students who fit into this model think that glaciers are masses of ice and/or snow that are constantly moving and flowing in response to the gravitational forces generated by their own weight (category 1A). This is in line with the glaciologist definition according to which a glacier is a dynamic body of ice in which ice itself moves from an area of accumulation to an area of loss (ablation) and, thus, involves the critical concept of glacier mass balance. In addition, all the students assigned to this model believe that the existence of any remnants of civilization at the bottom of the lake is impossible (category 5A), which is the exact opposite of what is claimed by religious legend and myth. During glacier development, students identify the involvement of several other combined processes (ablation, accumulation, transport, erosion, etc.), which would have resulted in the generation (and record) of two major glacial landforms (category 2A): depositional (e.g., glacial erratics, moraines) and erosional landforms (e.g., groves and striations, U-shaped valleys). The terminology used by most of these students (59.26%, n = 27) to name each landform is correct (subcategories 2A1 and 2A2). Furthermore, students that exhibit this model usually claim that glacier formation may have extended over long periods of time and a long time ago. In fact, most students (63.0%) are able to place the lake's formation in a precise time frame (3A). This fact, which could be considered anecdotal, is crucial to develop a scientific model of the lake's glacial origin because it involves an association of different physical and temporal concepts, and also contradicts the popular legend and tale.

Of the 9.7% out of the total sample (n = 279) who exhibited this explanatory mental model (Figure 1), 17 (63.0%) belonged to Group 2, 6 (22.2%) to Group 3, and 4 to Group 4 (14.8%). That is, no students in the pre-instructional group (Group 1) displayed this model, and its assignation decreases over time in the post-instructional group (GR2-GR3-GR4, Figure 1).

*b) Static mental model:* Most of the students assigned to this model (71.0%, n= 113) maintain that glaciers are static masses of ice that only freeze or melt (category 1C). Some of them suggest that this melting could be a consequence of warmer conditions brought about by climate change (1C2). The remaining 29.0% (n= 113) understand that there are several processes involved in glacier development, but do not mention glacial motion or transport (category B) and, thus, they are mostly unaware of the major erosional motion-associated properties of ice. Moreover, students who exhibit this mental model do not fully understand the scientific evidence (glacial landforms) employed to determine the lake's glacial origin (20.3% were allocated to category 2A, 42.5% to category 2B, and 37.2% to category 2C, n = 113).

This mental model is displayed by 113 students out of the total of 279 (40.5%) (Figure 1). From them, 23 (20.35%) belong to Group 1, 40 (35.40%) to Group 2, 26 (23.00%) to Group 3, and 24 (21.24%) to Group 4. Internal variation analysis of each group shows a weaker representation of this mental model in GR1 (20.0%, n = 85), and a regular presence of it in all the post-instructional groups (GR2, GR3 and GR4), which range from 52.0% to 40.0% (Figure 1).

*c) Catastrophic mental model*: all the students assigned to this category believe that glacier melting and the subsequent glacial lake formation are part of a relatively sudden and catastrophic event (category 5B). It can be inferred that students have a catastrophic view of surficial processes and the formation of the landscape. In this case, glacial melting would have led to the breakage of the Vega de Tera dam, flooding Ribadelago, and forming the Sanabria Lake. In fact, most of the students who exhibit this model (91.6%, n = 83) think that the water of the lake is so cold because it comes from the melting of a mass of ice (category 6C).

This mental model is displayed by 83 students out of the total sample of 279 (29.7%) (Figure 1): 45 (54.2%) are from Group 1, 13 (15.7%) are from Group 2, 6 (7.2%) are from Group 3, and 19 (22.9%) are from Group 4. Internal variation analysis for each group showed the highest representation of this mental model in GR1 (52.9%, n = 85), dropping dramatically in GR2 and GR3, and slightly increasing again in GR4 (31.7%, n = 60), (Figure 1).

*d) Naive mental model*: students attributed to this group display a poorly developed mental model and their ideas about the origin of the lake are vague and simplistic. Most of them think that its formation was related to something that occurred in a glacier or during or after a glaciation, but they

are not able to provide any accurate explanations. Others think that the lake's formation was related to processes that are, in themselves, abstract and vague, such as "tectonic movements." Almost half of the students (48.2%, n = 56) are unaware of any of the scientific evidence used by scientists to support the Sanabria Lake's glacial origin. In addition, 55% cannot provide an accurate temporal reference for its origin.

This mental model was exhibited by 56 students out of the total of 279 (20.1%) (Figure 1): 17 (30.3%) are from Group 1, 14 (25.0%) are from Group 2, 12 (21.5%) are from Group 3, and 13 (23.2%) are from Group 4. Internal variation analysis for each group shows that this mental model remains highly constant across all groups, which range from 20.2% (GR2) to 27.1% (GR1).

### 4.1. Statistical Analysis

A model-group comparison of the frequency of students' mental models was conducted by analyzing the differences between pre-intervention and post-intervention groups (GR1-GR2/GR3/GR4) and between individual groups.

The difference in the frequency of students' mental models between pre-intervention and post-intervention groups (GR1-GR2/GR3/GR4) proved significant [$\chi2$ (3, 279) = 39.29; $p < 0.00001 < 0.05$]. Thus, it can be concluded that fieldwork had a significant impact on mental model development. Likewise, the frequency of students' mental models between each individual pre-intervention and post-intervention group (GR1-GR2, GR1-GR3, and GR1-GR4) also showed significant differences ($\chi2$ (3, 169) = 39.53, $p < 0.00001 < 0.05$], $\chi2$ (3, 135) = 29.80, $p < 0.00001 < 0.05$], $\chi2$ (3, 145) = 11.14; $p = 0.011 < 0.05$], respectively).

When comparing the impact of intervention over time, the frequency of students' mental models between GR2-GR4 showed significant differences [$\chi2$ (3, 144) = 9.47, p = 0.023 < 0.05]. In turn, the difference in the frequencies of the students' mental models between GR2-GR3 and GR3-GR4 were non-significant ($\chi2$ (3, 134) = 2.49, p = 0.4757 > 0.05], $\chi2$ (3, 110) = 9.44; p = 0.092 > 0.05]). It can be concluded that fieldwork has a significantly different impact on the development of students' mental models depending on whether the instruction is recent or distant in time.

## 5. Discussion and Implications

The goal of this study was to examine pre-service teachers' mental models of glaciers and their effects in the formation of the landscape. Data collection was performed by means of an open-question questionnaire, which was completed by 279 university students. The results yielded four mental models. By identifying students' mental models before and after instruction, the study examined whether field trips could affect or modify such mental models. In contrast to previous research [39], the main results indicate that field trips positively affected the construction of students' mental models. Moreover, results show that some pre-service teachers were able to build more complex models after the educational field trip, which reinforces the idea shared by other authors that fieldwork has a significant impact on knowledge development [7]. This datum also agrees with other studies that have demonstrated the positive effect of instruction in mental model development [50,54,55].

In general terms, students' mental models about how glaciers function and how they create landforms prove relatively simplistic and incomplete. Furthermore, models include ideas and misconceptions that are similar to those found in previous research studies [25–28]: students are mostly unaware of the major erosional properties associated with glaciers and are ignorant of glacial dynamics. In addition, according to the present study, students do not realize that glaciers are bodies of ice that have a tendency to move downslope according to gravity.

Data indicate that fieldwork had a direct impact insofar as it helped students to reason about the processes involved in glacial landscape formation. In fact, the dynamic model (the most scientific) was only identified in post-instruction groups (GR2-4, Figure 1). Students who presented this mental model had a scientific view of glacier functioning and its effects: they understood that glaciers constantly move under the force of gravity and were aware of the erosive properties of ice. However, this view is partially incomplete, since students fail to elucidate all the components, concepts, and processes

involved, such as mass balance, thermal regime, or how glaciers flow. Furthermore, some of the students assigned to this model did not have an accurate temporal view of when glaciers were formed and, most importantly, they did not understand the duration of the erosive processes. This utter lack of ability to think across geological time in order to place correctly the Ice Age, connection between absolute ages, process duration, and occurrence has also been reported in previous studies focused on misconceptions about glaciers and on geological time [28–30], and also in other fields [56–58].

The static mental model is exhibited by several students, even in post-instruction groups (Figure 1). These students contemplate glaciers as static masses of ice that only freeze or melt, being, therefore, mostly unaware of the major erosional motion-associated properties of ice. This contrasts with previous studies [25] where 70% of the participating students thought that glaciers are masses of ice that move downslope.

The prevalence of the catastrophic mental model in GR1 students (52.9%, Figure 1) suggests that primary course pre-service teachers are strongly influenced by the historic event of the Ribadelago catastrophe. This idea strongly interferes with their mental reasoning about the formation of landscape features. Students who hold this mental model will need to learn more about the gradual and long-term erosional processes that can create landforms. They link the catastrophic flood to the origin of the lake. Likewise, they commonly associate the flooding with the rapid melting of the glacier, which would have caused the dam's breakdown. Similar flooding conceptions have been found in previous research studies [26]. In the context of the study, students provided an incomplete non-scientific explanation because they failed to contemplate the erosive processes that excavated the 55-meter-deep lake basin. This mental reasoning is reinforced and/or mixed with the religious myth of the village of Villaverde de Lucerna, which also attributes the lake's origin to a catastrophic event: a mystic flooding. In this case, the myth related to the presence of Jesus Christ in a village at the bottom of the lake is hard to believe. However, results show that such legends still live on in the minds of several pre-service teachers, who believe that the Sanabria Lake covers an ancient underwater village. Known by all the children of Zamora and told at the Sanabria Lake House Park through panels and videos, such a fable interferes with scientific reasoning about the formation of the lake and hinders the acquisition of scientific ideas. Throughout history, myths and legends have played a central role in the account of the origin of lakes and other geological processes, since, in the past, there were no scientific explanations for them [59]. In addition, floods are recurrent phenomena in the attempts of many legends and myths to explain geological processes [60]. In recent years, geomythology has studied whether there is any truth in these myths, and certain researchers have proved that some myths are based on age-old events, which has sometimes proved helpful to solve geological problems [60,61].

Data suggest that, after the educational field trip, some of the students seemed to organize a conceptual framework where new knowledge is placed at the top, above what they already knew. This means students' mental models acted as scaffolds to support future learning. Two years after instruction, some of the students (6.7%, n = 60, Figure 1) were able to construct a dynamic model to explain the lake's formation. It seems that students preserved an internal representation that, under new circumstances and even after a long time, gave rise to the building of a similar mental model to the one that had worked before and had been learned during fieldwork instruction. This could mean that learning is the result of a process that involves a set of episodic representations (mental models) that incorporate key and invariant elements into a more stable representation that operates in long-term memory. Such invariant elements appear again in the same way to shape the corresponding mental model [50].

Nevertheless, the presence of dynamic mental models decreases over time in post-instruction groups (from 16.7% in GR2 to 6.7% in GR4) and many other students exhibit mental models that dominate in GR1. This indicates that, even though knowledge could have developed through enrichment, key and generative elements (cognitive pieces of the mental model system), which most of the students added to their knowledge to construct their mental models, were not meaningful to them. In other words, it seems that fieldwork was not as efficient as expected for students to restructure

their concept of glaciers and their effects. Some authors devoted to other research fields contribute similar results and suggest that, if students only memorize scientific concepts that have no significance for them, they tend to forget such contents very quickly [62,63]. Therefore, the real challenge for educators is not only to teach science and learn how to build science models, but also to generate useful knowledge for students. In this context, the results also seem to reject the idea that people build mental models by reorganizing knowledge fragments [64], since the number of people with naive mental models remains unchanged with similar percentages across the four groups (Figure 1). In fact, naive mental models are believed to be built upon intuitive and fragmented pieces of knowledge [12]. Such disconnection between ideas and concepts likely thwarted the development of the scientific mental model. In order to understand scientific concepts and natural phenomena, students cannot rely on the simple memorization of concepts or the enrichment of their naïve theories. Instead, they need to be able to restructure their prior knowledge, which is based on everyday experience and lay culture, to facilitate the conceptual change [65]. Learning is not accumulating new knowledge or gap filling incomplete knowledge. Rather, learning is changing prior misconceived knowledge to science concepts and principles to be learned [66,67].

An analysis of the data shown in Figure 1 also suggests that, after instruction, many students changed their mental reasoning from: a) catastrophic to dynamic mental models or b) catastrophic to static mental models. In the first case, it seems that fieldwork led to the development of deep cognitive and conceptual learning. It seems that students were able to reconstruct their pre-instructional conceptual structures in order to allow the learning of the lake formation processes. The second case might be due to incomplete understating of the processes involved in the lake's formation and their times. The instruction taught students that the melting and retreat of a glacier is a relatively slow process, but they did not incorporate any information related to glacier dynamics. Even though the presence of the catastrophic mental model is rather low in GR2 and GR3, it reappears again in GR4 (31.7%). This likely indicates that, two years after the instruction, students go back to their primitive ideas to construct their mental models. In contrast to previous studies that suggest that the information learned during a field trip can be remembered for a long period of time [68], our results suggest that mental models constructed after fieldwork do not endure for a long time. In this case, students seem to have only retained in their minds that a sadly historic flooding occurred a long time ago, which is an idea that endorses and governs their mental reasoning and model construction.

The results obtained in this study provide a sound basis to make some educational recommendations for classroom instruction. The fact that the data indicate that fieldwork helped students to develop more complex mental models, even two years after the intervention, confirms that fieldwork had a positive impact on the teaching and learning of surficial processes and their effects on landscape formation. Therefore, from the present experience, it is highly recommended to use fieldtrips as part of science courses in teacher training studies at the university. However, this assumption must be taken with caution, because the results also show that this positive impact might not persist for a long time. The data also indicate that fieldwork alone does not necessarily mean that all students will acquire a deeper understanding. In this sense, research has demonstrated that cognitive learning from field trips can be enhanced by the use of pre-fieldwork and post-fieldwork activities in the classroom (see Reference [7] for a review). In fact, these activities might be crucial for conceptual learning, especially in light of the ephemeral nature of many of these field experiences. Furthermore, since it might be difficult for some students to learn how a physical feature is formed through fieldwork because the formation process cannot be directly witnessed, teachers should consider using other resources such as instantaneous simulation classroom activities or computer programs to help students develop a better understanding of geomorphological processes [69]. Likewise, models and modelling should play a crucial role in the teaching and learning of all scientific disciplines [13,70,71]. Additionally, this is especially important in Earth and Space Sciences, where there are numerous abstract concepts that are hard to understand from a human spatial or temporal perspective [72]. If educators do not work with modelling processes during their science lessons, students could face difficulties in building complex mental models, to the extent that some of them might not even manage to do so [73].

Teachers should also draw more attention to the fact that glaciers are masses of ice under constant movement, according to the force of gravity. As demonstrated in this paper, to assume that all students are aware of such processes and features is likely inadequate. Movement over extended periods of time and the strong erosional power of ice masses generate different erosional and depositional landforms. Hence, certain students may need help to develop an accurate temporal approach to understand the duration and functioning of the surficial processes involved in landscape formation. This would likely contribute to driving catastrophic views away. Nevertheless, the task is far from easy because natural processes occur at different time rates, and a certain degree of knowledge about the rate at which a process occurs is essential to assess its duration [74]. Thus, teachers should always bear in mind the numerous obstacles and limitations involved in the teaching and learning of any process or event related to geologic time [56,58,74,75]. The planning of prior and post-field work classroom activities dealing with the students' understanding of geologic time is also suggested since deep time and temporal scales are other major areas of misconceptions in Earth Sciences. Demonstrating for students that natural processes could occur in different ranges, from imperceptibly slow to exceptionally fast, could impact and improve their understanding of landscape formation.

**Author Contributions:** Conceptualization, Diego Corrochano and Alejandro Gómez-Gonçalves. Formal analysis, Diego Corrochano and Alejandro Gómez-Gonçalves. Funding acquisition, Diego Corrochano. Investigation, Diego Corrochano and Alejandro Gómez-Gonçalves. Methodology, Diego Corrochano and Alejandro Gómez-Gonçalves. Writing – original draft, Diego Corrochano and Alejandro Gómez-Gonçalves.

**Funding:** This research was funded by the project 18K117 of the University of Salamanca (Spain).

**Acknowledgments:** We would like to thank the two anonymous reviewers for their comments that improved the manuscript. We also thank all students who have participated in the study.

**Conflicts of Interest:** The authors declare no conflict of interest.

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
