# Peer review of "How Glaciers Function and How They Create Landforms: Testing the Effectiveness of Fieldwork on Students’ Mental Models—A Case Study from the Sanabria Lake (NW Spain)"

_geosciences, doi:10.3390/geosciences9050238_

Round 1
Reviewer 1 Report
Altogether, this manuscript deals with a neglected field within geoscience education research and contributes relevant results to this field.
My concerns are:
1) Line 90-100: It is said that only the publication of Happs (1982) would exist within the field of conceptual change research regarding glaciers. But Felzmann (2014, 2017) has published two papers dealing with students‘ conceptions about glaciers and ice ages. Czajka & McConnell (2018) have analysed students‘ ability to identify and explain an U-shaped valley. These results should be considered within 1.1. and 5. Also Trends research how students order geological events could be considered, because it reveals that „the Ice Age“ is a very badly understood geological time. This result could be used to interpret the answers in questions Q3 and Q4.
Czajka, C.D. & McConnell, D. (2018): An exploratory study examining undergraduate geology students’ conceptions related to geologic time and rates, Journal of Geoscience Education, DOI: 10.1080/10899995.2018.1480826
Felzmann, D. (2017): Students’ Conceptions of Glaciers and Ice Ages: Applying the Model of Educational Reconstruction to Improve Learning. Journal of Geoscience Education 65(3), 322–335.
Trend, R. (2001): An investigation into the understanding of geological time among 17-year-old students, with implications for the subject matter knowledge of future teachers. International Research in Geographical and Environmental Education, 10(3), 298–321.
2) It is not clear whether the students have to think from a glacial perspective when they answer questions 1 and 5. (Also it is not clear, whether the answers 1A1 and 1A2 refer only to glacial motion. Only the term „motion“ is mentioned.) The students of GR1 have no hint to refer to a glacial perspective when they have to think about the formation of Sanbira Lake. But when they try to answer question 2 they find the information of a glacial origin: Could they switch back to question 1 to change their first answer or did they have to finish question 1 before they were confronted with question 2? The mental models „static mental model“ and „catastrophic mental model“ are described as if they only refer to glacial perspectives. But what about answers that describe a catastrophic flood caused by other sources than glacial melt water (e.g. rainfall, melted snow)? The model „static mental model“ refers to the question how a glacier is structured (= as a static mass of ice), but the model „catastrophic mental model“ refers to the question how fast a specific landform (Sanabira Lake) is formed. According to my understanding, these two models are not mutually exclusive to each other, because they deal with different aspects. For example, a catastrophic flood could result from a sudden melting of a static mass of ice which is understood as a „glacier“. The mental model „catastrophic mental model“ is based on the answer 5B. But why could a student not think about Sanabria Lake forming as a catastrophic event while he understands a glacier as a constant flow of ice? Therefore it is recommended to differ between mental models that refer to the question of the time structure of geomorphological formations, e.g. „static“ („has always looked like today“) – „catastrophic“ – „gradualistic“, and mental models that refer to the question of the structure of a glacier, e.g. "static" – "dynamic". Actually, the mental models confound the two aspects "structure/function of a glacier" and "time structure of geomorphological formation".
3) Are there further possible sources which may affect the understanding of the Sanabria Lake and glaciers in general while studying at the university? Is the described excursion the only course dealing with geomorpholgy/geoscience/glaciers during the participants‘ studies at university? The „development“ of the mental models from GR1 to GR4 is interpreted as if there are is no further input about such themes. Therefore it is recommended to give more information about the participants and their studies at this university.
4) The research is embedded within conceptual change research. The only reference to a conceptual change theory is the one to Chi, Slotta & De Leeuw (Line 66-75). Why only this one? Can the results of the research, especially the „development“ of the mental models, be explained by a shift of the ontological category? This point should be discussed in the discussion.
5) Why is a cluster analysis, based on the categorised answers in table 1 and 2, not conducted to construct the different mental models? Maybe there are some causes for not using this way which I do not know, because I am not an expert in using such methods…
Author Response
We warmly appreciate your supportive comments that strongly improved our work. Most of your comments and suggestions have been considered and clarified in the reviewed version of the text. In response to your comments:
1) Line 90-100: It is said that only the publication of Happs (1982) would exist within the field of conceptual change research regarding glaciers. But Felzmann (2014, 2017) has published two papers dealing with students‘ conceptions about glaciers and ice ages. Czajka & McConnell (2018) have analysed students‘ ability to identify and explain an U-shaped valley. These results should be considered within 1.1. and 5. Also Trends research how students order geological events could be considered, because it reveals that „the Ice Age“ is a very badly understood geological time. This result could be used to interpret the answers in questions Q3 and Q4.
Czajka, C.D. & McConnell, D. (2018): An exploratory study examining undergraduate geology students’ conceptions related to geologic time and rates, Journal of Geoscience Education, DOI: 10.1080/10899995.2018.1480826
Felzmann, D. (2017): Students’ Conceptions of Glaciers and Ice Ages: Applying the Model of Educational Reconstruction to Improve Learning. Journal of Geoscience Education 65(3), 322–335.
Trend, R. (2001): An investigation into the understanding of geological time among 17-year-old students, with implications for the subject matter knowledge of future teachers. International Research in Geographical and Environmental Education, 10(3), 298–321.
Authors: In the reviewed version of the manuscript, we have incorporated all of these references. As you suggested, we have also modified the introduction and the discussion epigraphs (1.1 and 5). Thank you very much for these references that have improved the introduction of our paper. We are extremely grateful to this suggestion.
2) It is not clear whether the students have to think from a glacial perspective when they answer questions 1 and 5.
Authors: It is true that in the earlier draft of the manuscript it is not clear whether the students have to think from a glacial perspective when they answered the questionnaire. That was happened because the introductory heading of the survey, where we explained to the students that the Sanabria Lake was of glacial origin, was not incorporated in the text due to an oversight. Now an explanatory sentence has been incorporated in epigraph 3.2. Thanks for the advice.
Also it is not clear, whether the answers 1A1 and 1A2 refer only to glacial motion. Only the term „motion“ is mentioned.
Authors: The adjective “glacial” has been added and now the term “glacial motion” is employed. Please, check Table 1 and text (epigraph 4).
The students of GR1 have no hint to refer to a glacial perspective when they have to think about the formation of Sanbira Lake. But when they try to answer question 2 they find the information of a glacial origin: Could they switch back to question 1 to change their first answer or did they have to finish question 1 before they were confronted with question 2?
Authors: As mentioned above, we have incorporated an explanatory sentence in epigraph 3.2 explaining that in the introductory paragraph of the questionnaire we explained to the students that the Sanabria Lake was of glacial origin. Therefore, students of GR1 could think from a glacial perspective when they answered the questionnaire. In addition, we also have clarified that we did not established time-limit to answer the questionnaire and “during this time, students were able to switch back to previous questions to change their answers if they wished” (epigraph 3.2).
The mental models „static mental model“ and „catastrophic mental model“ are described as if they only refer to glacial perspectives. But what about answers that describe a catastrophic flood caused by other sources than glacial melt water (e.g. rainfall, melted snow)?
Authors: Please, note that as indicated above, students answered the questionnaire knowing that the Sanabria Lake was of glacial origin (this is highlighted in the introduction of the questionnaire and in the question 2). Therefore, all answers were interpreted from a glacial perspective. Indeed, in several cases, students who held the catastrophic mental model explicitly explained that a flood that resulted from glacial melting caused the breakdown of the Vega del Tera dam.
The model „static mental model“ refers to the question how a glacier is structured (= as a static mass of ice), but the model „catastrophic mental model“ refers to the question how fast a specific landform (Sanabira Lake) is formed. According to my understanding, these two models are not mutually exclusive to each other, because they deal with different aspects. For example, a catastrophic flood could result from a sudden melting of a static mass of ice which is understood as a „glacier“. The mental model „catastrophic mental model“ is based on the answer 5B. But why could a student not think about Sanabria Lake forming as a catastrophic event while he understands a glacier as a constant flow of ice? Therefore it is recommended to differ between mental models that refer to the question of the time structure of geomorphological formations, e.g. „static“ („has always looked like today“) – „catastrophic“ – „gradualistic“, and mental models that refer to the question of the structure of a glacier, e.g. "static" – "dynamic". Actually, the mental models confound the two aspects "structure/function of a glacier" and "time structure of geomorphological formation".
Authors: It is true that all the mental models defined in the present study (and in many other research papers in the literature) might be not mutually exclusive to each other strictly speaking. But this could be motivated because mental models are individual, dynamic and abstract artefacts of reasoning and thus, it is difficult when we try to put them into a pigeonhole.
In the present paper, the “static mental model” does not refer to the internal structure of a glacier or glacial dynamics. Instead, it highlights the fact that many students do not think about how the erosion of the lake basin could have been produced without the existence of glacial movement (and its associated erosion). See, for example, lines 433-435. In order to clarify the characteristics of this model the sentence “and thus, they are mostly unaware of the major erosional motion-associated properties of ice” has been added to the description (4b, results, static mental model).
In turn, the catastrophic mental refers to the fact that students do not think from a “deep time perspective”, and they are not aware of the long time needed to form a glacial lake. That is, all defined mental models are about the processes (and their timings) involved in the landform formation, not into the own glacial functioning and glacial dynamics properly said.
3) Are there further possible sources which may affect the understanding of the Sanabria Lake and glaciers in general while studying at the university? Is the described excursion the only course dealing with geomorpholgy/geoscience/glaciers during the participants‘ studies at university? The „development“ of the mental models from GR1 to GR4 is interpreted as if there are is no further input about such themes. Therefore it is recommended to give more information about the participants and their studies at this university.
Authors: Participants have not received further instruction about glaciers/geomorphology/Earth Sciences after their mentioned second course subjects and associated fieldtrip. As you suggested, this has been clarified in the text and we have included further information about the participants and their studies at the university in the epigraph 3.3.
4) The research is embedded within conceptual change research. The only reference to a conceptual change theory is the one to Chi, Slotta & De Leeuw (Line 66-75). Why only this one? Can the results of the research, especially the „development“ of the mental models, be explained by a shift of the ontological category? This point should be discussed in the discussion.
Authors: In the present paper we have not developed an ontological categorization of students’ ideas, so we think that the development of the mental models should not be necessary explained according to the Chi’s ontological theory. In fact, research on students’ conceptions and conceptual change has been embedded in various theoretical frames over the past three decades. Thus, conceptual change could be also explained with other widely accepted theories such as the Vosniadou’s framework theory (Vosniadou & Brewer, 1994) or diSessa’s knowledge in pieces theory (1988). Because the main aim of this paper was focused on the (long-term) effectiveness of fieldwork in the learning process, rather to analyse these ways of understating the conceptual change, we have preferred to not analyse this approaches in deep. However, from your suggestions, we have tried to be more emphatic highlighting the importance of the conceptual change in the learning process and some explicit references have been added in the introduction and discussion epigraphs. Thank you for your comment that we will take into account in future researches.
5) Why is a cluster analysis, based on the categorised answers in table 1 and 2, not conducted to construct the different mental models? Maybe there are some causes for not using this way which I do not know, because I am not an expert in using such methods…
Authors: As early mentioned, the idiosyncratic nature of mental models makes difficult to put them into a pigeonhole: although mental models are useful for individuals, they are also dynamic, incomplete, imprecise and sometimes incoherent. Thus, the diagnosis of clear mental models is difficult and may be performed in different ways. In the present paper, the questions about how the lake was formed and which processes intervened in its formation (question 1, Table 1), and about if could there be a submerged village at the bottom of the Sanabria Lake (question 5, Table 2) are considered keys to discover students’ understating and reasoning processes on the issue of the Sanabria lake formation. That is, the questions of the questionnaire have a kind of hierarchy when trying to define the mental models. The information of the other answers (questions 2, 3, 4 and 6) was used to complete and to check the internal coherence of mental models. Taking into account this clarification, cluster analysis encompassing all the categorised answers is not suggested. However, in the results epigraph (lines 317-384), we have tried to explain in detail the different categorised answers that students gave in each question. Finally, in order to clarify the employed method in mental model definition, an additional explanation has been added (lines 312-313).
Yours sincerely,
Diego Corrochano Fernández (dcf@usal.es)
Alejandro Gómez-Gonçalves (algomez@usal.es)

Reviewer 2 Report
It was vital that you recommended the continuation of in-class modeling and simulation after field experiences. A field trip/field work is highly valuable but I'm not sure if you could be more emphatic about the follow up so as not to resort to previously held conceptions/misconceptions, rather than an evolution of their mental model and demonstrate true understanding. The other aspect relates to the temporal scale - and the presumption that the pre-service teachers are comfortable and accepting and understand to long periods of time it takes for glaciers to create a landscape. An extension activity (perhaps prior to and post field work) that entertains their understanding of geologic time is important as that is another major area of misconceptions in the earth sciences. Demonstrating for pre-service teachers and young learners that geologic processes occur at both ends of the temporal scale could impact their understanding and appreciation of geology. I am curious of your subjects - especially those that chose no answer or wrote they had no sense of anything - could they truly even "see" what they were looking at or looking for? The other aspect to follow up on is - beyond their understanding - is their affective behavior - how has the field experience adjusted (if it did) their appreciation of Earth and its processes and the beauty of the landscape. And - as pre-service teachers - do they now understanding the importance of real-world scientific experiences for their own students?
Also - do you recommend that - in the course of the university education - that their science experience - either in a science content class or a science education methods class - infuse a field trip/field work - as part of the requirement for the course?
Author Response
We warmly appreciate your supportive comments that strongly improved our work. Most of your comments and suggestions have been considered and clarified in the reviewed version of the text. In response to your comments:
It was vital that you recommended the continuation of in-class modeling and simulation after field experiences. A field trip/field work is highly valuable but I'm not sure if you could be more emphatic about the follow up so as not to resort to previously held conceptions/misconceptions, rather than an evolution of their mental model and demonstrate true understanding.
Authors: In class-modelling and simulation activities after fieldwork were suggested in the earlier draft of the manuscript. However, as you suggested, we have tried to be more emphatic highlighting the importance of the follow-up activities (please, see discussion epigraph). We have added some references and mentioned some methods, such as conducting teaching experiments after fieldwork that can improve the conceptual development of students (Felzmann, 2014).
The other aspect relates to the temporal scale - and the presumption that the pre-service teachers are comfortable and accepting and understand to long periods of time it takes for glaciers to create a landscape. An extension activity (perhaps prior to and post field work) that entertains their understanding of geologic time is important as that is another major area of misconceptions in the earth sciences. Demonstrating for pre-service teachers and young learners that geologic processes occur at both ends of the temporal scale could impact their understanding and appreciation of geology.
Authors: In the reviewed version of the manuscript, we have incorporated some references associated with the importance to master the main events across geological time in order to understand the Ice Age and how glacier landforms are created. We have modified the introduction (1.1) and the discussion epigraphs (5). Moreover, we also have added a suggestion to plan some pre/post classroom activities dealing with deep time (lines 532-535). Thank you very much for the suggestion.
I am curious of your subjects - especially those that chose no answer or wrote they had no sense of anything - could they truly even "see" what they were looking at or looking for? The other aspect to follow up on is - beyond their understanding - is their affective behavior - how has the field experience adjusted (if it did) their appreciation of Earth and its processes and the beauty of the landscape. And - as pre-service teachers - do they now understanding the importance of real-world scientific experiences for their own students?
Authors: These questions are very interesting but, unfortunately, we have no data to answer them in an accurately way. We take those suggestions into account in order to include affective and professional items in our future researches.
Also - do you recommend that - in the course of the university education - that their science experience - either in a science content class or a science education methods class - infuse a field trip/field work - as part of the requirement for the course?
Authors: We recommend vividly, from our experience, the use of field trips as part of science course due to the fact that this method can improve the students’ experience from an affective and a cognitive side. We have added a sentence where it is said in an explicit manner (lines 506-507).
Again, thank you very much for all your critical and supportive comments, which are warmly appreciated.
Yours sincerely,
Diego Corrochano Fernández (dcf@usal.es)
Alejandro Gómez-Gonçalves (algomez@usal.es)

Round 2
Reviewer 1 Report
Altogether, the authors have incorporated my concerns in a reasonable and transparent way into their new version. Their answers to my comments are extensive and comprehensible. Thank you very much. Especially the background information the students had during writing their answers are much better understood. In this way the interpretations of their answers are more comprehensible.
The mentioned literature (Felzmann, Czjaka, Trend) is taken into account only a little. One sentence (537-538) referring to this literature is wrong: "Teaching experiments" are not a method of teaching (for example after a fieldtrip) but a method of conceptual change research to explore students' conceptual development. It is recommended to cancel this sentence.
Author Response
RESPONSES FOR REVIEWER 1
Thank you very much again for all your supportive and constructive suggestions. In response to your comments:
Altogether, the authors have incorporated my concerns in a reasonable and transparent way into their new version. Their answers to my comments are extensive and comprehensible. Thank you very much. Especially the background information the students had during writing their answers are much better understood. In this way the interpretations of their answers are more comprehensible.
The mentioned literature (Felzmann, Czjaka, Trend) is taken into account only a little. One sentence (537-538) referring to this literature is wrong: "Teaching experiments" are not a method of teaching (for example after a fieldtrip) but a method of conceptual change research to explore students' conceptual development. It is recommended to cancel this sentence.
Authors: As you suggested, we have deleted the sentence of lines 537-538. We have also slightly modified the abstract and the introduction, and now we give a little more information of the cited literature about glaciers and misconceptions (lines 106-113). Additionally, we also have added a reference (Felzmann 2017) in the discussion epigraph about glaciers and flooding conceptions. We hope all of these changes are appropriated.
Thank you very much again.
Diego Corrochano Fernández.